# Molecular Genomic Analyses of *Enterococcus cecorum* from Sepsis Outbreaks in Broilers

**DOI:** 10.3390/microorganisms12020250

**Published:** 2024-01-25

**Authors:** Douglas D. Rhoads, Jeff Pummill, Adnan Ali Khalaf Alrubaye

**Affiliations:** 1Program in Cell and Molecular Biology, University of Arkansas, Fayetteville, AR 72701, USA; jpummil@uark.edu (J.P.); aakhalaf@uark.edu (A.A.K.A.); 2Department of Biological Sciences, University of Arkansas, Fayetteville, AR 72701, USA; 3Arkansas High Performance Computing Center, University of Arkansas, Fayetteville, AR 72701, USA; 4Department of Poultry Science, University of Arkansas, Fayetteville, AR 72701, USA

**Keywords:** broiler, Enterococcus, sepsis, osteomyelitis, phylogenomics, core genome mutations, pathogenesis

## Abstract

Extensive genomic analyses of *Enterococcus cecorum* isolates from sepsis outbreaks in broilers suggest a polyphyletic origin, likely arising from core genome mutations rather than gene acquisition. This species is a normal intestinal flora of avian species with particular isolates associated with osteomyelitis. More recently, this species has been associated with sepsis outbreaks affecting broilers during the first 3 weeks post-hatch. Understanding the genetic and management basis of this new phenotype is critical for developing strategies to mitigate this emerging problem. Phylogenomic analyses of 227 genomes suggest that sepsis isolates are polyphyletic and closely related to both commensal and osteomyelitis isolate genomes. Pangenome analyses detect no gene acquisitions that distinguish all the sepsis isolates. Core genome single nucleotide polymorphism analyses have identified a number of mutations, affecting the protein-coding sequences, that are enriched in sepsis isolates. The analysis of the protein substitutions supports the mutational origins of sepsis isolates.

## 1. Introduction

Enterococci are Gram-positive firmicutes comprising commensals and pathogens in a wide range of vertebrates and plants [1,2], and are prevalent in animal gastrointestinal tracts [1,2,3,4], nature [1,5], and even food processing systems [6]. Some species are found in diverse vertebrates, such as *Enterococcus faecalis*, *Enterococcus faecium*, and *Enterococcus avium*, which have been isolated from both mammals and aves [1,2]. In humans, these species have been associated with infections of the urinary tract, bloodstream, heart, and central nervous system [7,8]. Some Enterococcal species appear to be restricted to particular host species or classes [2]. *Enterococcus cecorum* is primarily associated with avian species and has been isolated from the intestines or cloaca of galliforms and neoornithes [1,2,9]. Comparative genomics based on the Enterococcus core genome places *E. cecorum* most closely related to *Enterococcus columbae* (host birds) within a clade associated with plant, aquatic, and bird hosts [2]. *E. cecorum* typically colonizes the intestines of chickens at around 3 weeks of age [10,11,12]. Pathogenic strains have been associated with osteomyelitis affecting the flexible thoracic vertebrae (kinky back) and the proximal femoral (femoral head necrosis) or tibial (tibial head necrosis) growth plates. Borst et al. [13] identified a cluster of twelve genes that appear to distinguish osteomyelitis isolates from commensals, likely to be involved in virulence via capsular modifications. The evolution of pathogenesis and drug resistance in *E. faecalis* and *E. faecium* has been associated with mobile elements and the acquisition of capsular gene clusters [14,15]. Phylogenomic analysis of more than 100 genomes of *E. cecorum* isolates from French broilers suggested that a specific lineage of pathogenic isolates is infecting broilers in Europe and the United States [16].

The recent occurrence of sepsis outbreaks in broilers caused by *E. cecorum* is characterized by onset during the first few weeks post-hatch, prior to the normal colonization of the intestinal tract [17]. However, there are older reports of sepsis in broilers caused by *E. cecorum*, with losses of 5–8%, characterized by pericarditis, bacteremia, and osteomyelitis [18]. The mechanisms of the horizontal or vertical spread of *E. cecorum* in broiler flocks are not known, although particular facilities or flocks may experience repetitive outbreaks [17,18]. Recent outbreaks may be more problematic with the removal of early administration of antibiotic growth promoters in flocks [17,19,20].

The questions we wished to address included whether specific mutations or gene acquisition are driving recent *E. cecorum* septicemia outbreaks in broilers, whether sepsis and osteomyelitis isolates are distinct lineages, and whether high-resolution bacterial genomics can inform responses to sepsis outbreaks. To answer these questions, we assembled genomes from recent isolates from corporate surveys during sepsis outbreaks. These assemblies and all publicly available genomes were then analyzed to determine the whole-genome relatedness of pathogenic and non-pathogenic isolates. The assemblies were compared to determine whether specific genes or gene clusters were essential or highly associated with disease traits (e.g., pathogen vs. commensal, or osteomyelitis vs. sepsis). Similarly, we analyzed whether specific mutations in the core genes are associated with disease traits. These findings are critical for guiding the development of new treatments, vaccines, and management strategies for mitigating sepsis outbreaks.

## 2. Materials and Methods

### 2.1. New Genome Assemblies

Cultures were grown in tryptic soy broth (Difco) +5% chicken serum (Gibco ThermoFisher, Waltham, MA, USA) in 5% CO_2_, and archived in 40% glycerol at −80 °C. DNA was isolated from 20 mL of overnight stationary-phase broth cultures by treatment with lysozyme, followed by standard SDS-lysis, organic extraction, RNAse/protease digestion, and ethanol precipitation [21,22]. The final purification was performed using NanoSep 100 k Omega spin cups according to the manufacturer’s instructions (Pall Corporation, Port Washington, NY, USA). Purified total bacterial DNAs were submitted to SeqCenter (Pittsburgh, PA, USA) for 2 × 151 paired-end sequencing on an Illumina NextSeq 2000 (Illumina Corp., San Diego, CA, USA). Paired-end 2 × 251 sequence data for 22 additional isolates were obtained from the laboratory of Luke Borst (North Carolina State University). Sequence data were uploaded to BV-BRC [23] for processing through the Trim Galore pipeline and Unicycler genome assembly.

Whole-genome sequence reads and assemblies were submitted to NCBI under BioProject PRJNA1050746, which included BioSamples SAMN38750234 through SAMN38750267 and accessions JAXOGD000000000 through JAXOHK000000000.

### 2.2. Genome Analyses

Assemblies were analyzed for comparative genomics using Proksee [24] and for the location of mobile element genes [25]. Genomic islands were identified using IslandViewer 4 [26]. Prophages were identified using Phaster [27]. NCBI genome assemblies were downloaded 11 February 2023, with genome_updater v0.6.3 (https://github.com/pirovc/genome_updater). The phylogenomic trees of genome assemblies were produced based on kmer comparisons using PopPUNK v2.6.0 [28] and based on core genome single nucleotide polymorphisms (SNPs) using ParSNP v1.7.4 [29]. Newick trees were midpoint rooted using Archeopteryx v 0.9928 beta [30] and rendered in MicroReact [31]. Assemblies were annotated using Prokka v1.14.5 [32]. Pan and core gene partitioning was performed using Roary v3.13.0 [33]. Scoary v1.6.16 [34] was used for the pan-genome-wide association. ParSNP and Maast v1.0.8 [35] were used to generate vcf for core genome SNPs, which were then processed further in Microsoft Excel to identify possible trait-associated SNPs. Specific polypeptide sequences were extracted from the Prokka annotation files based on description text using the Linux grep command. Fasta files were converted to and from .tab format using a Biopython script (https://sequenceconversion.bugaco.com/converter/biology/sequences/fasta_to_tab.php; accessed September 2023). Excel was used to curate .tab formatted sequences to filter for evident orthologs based on amino and terminal sequences, and for full-length orthologs. For polypeptide variant identification, the sequences were aligned using Clustal W in MegAlignPro (Lasergene v17 (DNAStar, Inc.; Madison, WI, USA) and variation tables exported to Microsoft Excel. Signal sequences and cleavage sites were predicted using SignalP-6.0 [36].

## 3. Results and Discussion

### 3.1. Genome Assemblies from Sepsis Surveys

*Enterococcus cecorum* isolates from commercial operation samplings were obtained from poultry diagnostic laboratories representing samplings from 2020 to 2021 (Table 1). These included environmental samples as well as necropsy samples, including internal organs. We also included two bacterial chondronecrosis with osteomyelitis (BCO) isolates from our own sampling of commercial broiler farms [22] and our poultry research farm (UAPRF). The total DNA was purified from each isolate and submitted for Illumina 2 × 151 sequencing. The reads were trimmed and assembled as described in the Materials and Methods section. The assemblies (Table 1) ranged from 32 to 105 contigs with an average of 62 contigs. The total assembled base pairs ranged from 2.15 to 2.49 Mbp with an average of 2.27. The N50 values ranged from 49.7 to 209.1 kbp with an average of 108.9.

### 3.2. Considerations Regarding Disease Trait Classifications

A major issue in any of these analyses is knowing the phenotypic traits of each isolate regarding virulence/pathogenicity. The BioSample data from NCBI may not clearly identify the host disease state or the actual anatomical source of the culture. For example, if an isolate was obtained from the bone marrow, should it be classified as a case of sepsis or BCO? If an isolate was obtained from hatchery egg waste from an infected breeder flock, the virulence potential of that isolate is unknown. Therefore, we made every effort to be strict in our interpretation (see legend to Figure 1) as to whether an isolate was from BCO (osteomyelitis or bone marrow) or from sepsis (peritoneum, heart, liver, and blood). For the purposes of our analyses, we reserved the sepsis isolates as those isolated from an internal organ or compartment, whereas the BCO isolates were from bone marrow or infected joints. We also defined a chicken disease group (145 genomes) as BCO isolates plus the strictly defined sepsis isolates, which excludes isolates from the air sac, egg waste, environmental swabs, intestines, cecum, and unspecified clinical samples. All genomes not in the chicken disease group were assigned to the non-phenotypic group. Strain assignments to the phenotypic groups are listed in Appendix A.

### 3.3. Phylogenomic Analysis of Sepsis and Osteomyelitis Isolates

These 34 new assemblies along with 193 *E. cecorum* genomes that had been deposited in NCBI were used to generate phylogenomic trees to discern the relationships and possible origins of the BCO and sepsis isolates. Phylogenomic trees were generated for all 227 genomes using either kmer genome comparisons using PopPUNK (Figure 1) or shared core genome SNPs using ParSNP (Appendix A). Although the branch lengths were different for the two methods, the overall topology was similar, and clustering was very similar. Regardless of using a strict or relaxed definition of BCO or sepsis the phylogenomic analyses all support a polyphyletic origin for both the BCO and sepsis isolates. Red circles for BCO appear in all branches, and the green triangles/circles for chicken sepsis appear in multiple locations. There was one large cluster (A) of US sepsis isolates that was related to a BCO isolate we obtained from tibial pus from a broiler in 2016 (1415). Within this cluster were two vertebral osteomyelitis isolates (LB5924 and LB5925). Cluster B contained LB5800, a sepsis isolate from the heart, four closely related isolates from egg transfer residues (LB5857, LB 5916, and LB5876), an isolate from a case of air-saacculitis (LB5880), two distantly related isolates, a BCO femoral head necrosis isolate (1675), and an intestinal isolate (LB5923). Cluster C contained one sepsis isolate (1750) and two vertebral osteomyelitis isolates (LB5836 and LB5843). Cluster D contained no evidence of BCO or sepsis isolates as it consisted of isolates from cull eggs, or egg transfer residue, along with one isolate (LB5922) with an incomplete documentation of isolation source. The tree contained 100 genomes from isolates collected from 16 poultry farms in France between 2007 and 2017. Some of these genomes were from probable sepsis cases (red circles), with many representing pericarditis isolates [16]. These isolates appeared throughout the tree, with clusters including BCO and probable sepsis virulence phenotypes. There were also six clinical isolates from humans (green plus nodes, Figure 1 and Appendix A) that were distributed among the chicken isolates.

**Figure 1 microorganisms-12-00250-f001:**
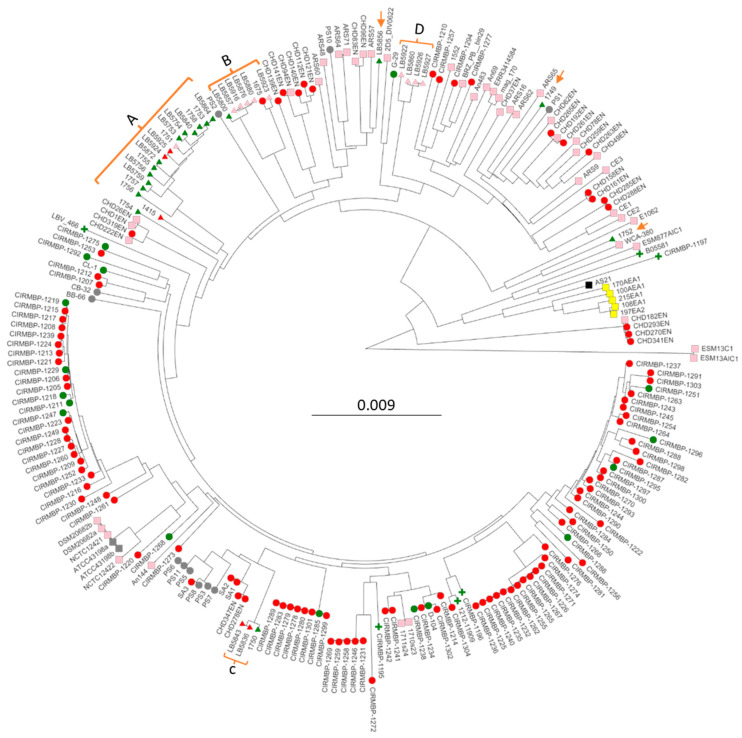
PopPUNK phylogenomic tree of 227 *Enterococcus cecorum* isolates. Strain names are the leaves. Clusters of USA isolates from the sepsis survey are bracketed with letters denoting clusters (Table 1), with orange arrows indicating solo isolates. Node colors indicate isolation anatomical location: pink—air sac, intestine, cloaca, feces, egg residue, feces, rinsate, and swab; green-blood—heart, liver, spleen, and peritoneum; red—air-sacculitis, bone marrow, leg joint, spine, and osteomyelitis; gray—unknown; yellow—meat; black—reactor. Node shape indicates the isolate source: circle—UA poultry research farm, osteomyelitis, and research clinic; triangle—corporate survey; plus—human; square—not available/unknown.

Of note, the phylogenomic analyses also included cloacal isolates from Harris Hawk (ESM13C1 and ESM13AIC1), black vulture (E1062, ESM877AIC1), pig feces (WCA-380), and internal organs from ducks (D-104) and goose (G-29). The Harris Hawk isolates appeared to be the most distant from the other 225 isolates. The goose G-29 isolate was located within the branches leading to cluster D. The pig and one of the black vulture isolates cluster with a solitary USA sepsis isolate, 1752. The duck internal organ/sepsis isolate D-104, isolated in Poland, was placed in a cluster with French sepsis and BCO isolates.

We interpret all of the phylogenomic data as most consistent with pathogenic isolates arising from non-pathogenic isolates and that sepsis isolates are closely related to BCO isolates. Our conclusions contrast with those from previous analyses of smaller collections of genomes that suggested distinct lineages for commensal and pathogenic isolates of *E. cecorum* [13,16,37]. 

### 3.4. Is Gene Acquisition Central to Sepsis or Disease Traits?

If osteomyelitis and/or sepsis are polyphyletic then this could be driven by gene acquisition and/or by mutation. Based on a comparison of the genomes of three BCO isolates with those of three cecal isolates, Borst et al. [13] proposed that a cluster of 12 genes possibly related to capsular modification distinguished the BCO isolates from the cecal isolates. Using the SA1 assembly of Borst as our reference genome, the Prokka annotation places the 12 gene cluster as protein-encoded genes (PEG) 312 through 323. We used BLASTp to search the Prokka-predicted polypeptides from all 227 *E. cecorum* genomes for these 12 polypeptides. BLASTp results were tabulated by combining values for percent identity and query coverage (pid × qcov/100) and averaged across all genomes according to the phenotypic group: none, chicken disease, BCO, strict sepsis (summarized in Table 2 with complete results in Appendix A). The results show that the cluster of capsular genes is conserved in approximately 70–80% of the isolates classified as chicken disease, and for those classified as BCO, the conservation may be higher in the strict sepsis group, but the number of those isolates is significantly lower (34 vs. 113). Additionally, the presence of these capsular genes in the isolates within the none group (no disease or disease association not known) is around 30 to 40% of the isolates.

Unicycler assemblies of our 34 assemblies were surveyed for circular elements. This identified three potential plasmids in isolate 1754. We used BLASTn searches of all 227 genomes to determine how widespread these contigs are as a signal for horizontal genetic exchange. Contig 25 (6237 bp) from 1745 appears to be more than 75% conserved in 24 assemblies: 1415, 1754, An144, ARS48, ARS60, ARS65, CHD182EN, CHD270EN, CHD278EN, CHD341EN, CHD347EN, CHD83EN, CHD96EN, CIRMBP-1218, CIRMBP-1233, CIRMBP-1259, CIRMBP-1276, CIRMBP-1282, CIRMBP-1284, CIRMBP-1285, CIRMBP-1294, PS10, PS2, and SA2. Contig 27 (4525 bp) is more than 75% conserved in 10 assemblies: 1415, 1754, An144, ARS60, CHD83EN, CHD96EN, CIRMBP-1234, CIRMBP-1238, CIRMBP-1273, and PS2. Contig 31 (2418 bp) is more than 75% conserved in only the 1754 genome; the closest related contig is in An144 and only 71% is conserved. The isolates containing homologous contigs are scattered around the phylogenomic tree (Figure 1) but are sometimes found in highly similar genomes, indicative of both horizontal transfer and shared origins. Only some genomes containing contig 25 homologs contain contig 27 homologs (and vice versa). Analysis of contig 25 using Prokka annotation and BLASTp of NCBI NonRedundant indicates six predicted CDS that annotate as four hypothetical proteins, a replication RepB initiation protein, and a plasmid recombination protein. Similarly, for contig 27, there are four annotated CDS: a hypothetical protein, a plasmid recombination protein, a CopG transcriptional regulator, and a replication RepB protein. Contig 31 annotates as three CDS: a hypothetical protein, a transcriptional regulator, and a replication RebB protein. Therefore, mobile plasmids appear to exist in *E. cecorum* but do not seem to be widespread or to be drivers of the emergence of sepsis. However, this is only based on the identification of circular elements from the Unicycler assemblies in the 34 genomes we have generated.

Our Prokka–Roary analysis of 227 genomes suggests that the core + pan-genome of this species contains 14,728 genes with 878 genes in the strict-core genome (99% ≤ strains ≤ 100%), 335 genes in the soft-core genome (95% ≤ strains < 99%), 1689 genes in the shell genome (15% ≤ strains < 95%), and 11,826 cloud genes (strains < 15%). Prior analysis of 140 genomes suggested a pangenome of 8523 gene clusters with 1207 genes in the strict-core genome, 4664 genes in the accessory genome, and a unique genome of 2652 [16]. Thus, our results appear to expand the pangenome by over 6000 genes with similar gene content for the sum of the strict-core and soft-core to the previous numbers for a strict-core genome.

Scoary analyses of the pan-genome with respect to phenotypic traits (Appendix A) were employed to determine whether any particular loci or gene clusters had high specificity for either chicken disease isolates or for strict sepsis. We also analyzed the chicken disease trait, which compared BCO isolates, along with the sepsis isolates, to the isolates for which there is no known disease phenotype. Since many of the isolates for which there is no known disease phenotype may actually include pathogenic isolates, we were skeptical of any meaningful results for this particular comparison. The data for loci positively or negatively associated with Chicken Disease identified 195 genes with corrected *p*-values < 0.05 (Appendix A). Of the 159 loci positively associated (Odds ratio ≥ 5) with chicken disease trait, 63 were annotated as something other than a hypothetical protein. Of the 20 loci negatively associated (Odds ratio ≤ 0.2), only 6 were annotated as something other than a hypothetical protein. For these 159 loci, we identified 12 clusters of two or more loci based on their position in the genomes of either of the three BCO isolates. Seven of the locus clusters were present and clustered in all three isolated genomes examined (Appendix A). Functions associated with these clusters include glycosylation loci, including the capsular genes identified by Borst [13], carbohydrate/sugar transferase systems, probable transposable elements, and vitamin/biotin transferase systems. When we analyzed all the genomes for loci associated with the strict sepsis group, the data identified 101 loci with 97 positively associated and 4 negatively associated (Appendix A) However, of these, loci 94 were apparently identified because they are primarily found in in the genomes present in cluster A in Figure 1. To reduce the bias of Cluster A, we reduced the representation from that cluster by excluding some of the Cluster A isolates to generate the trait group Sepsis Strict Reduced (Appendix A). The Scoary output using Sepsis Strict Reduced indicated that there were no loci with Bonferonni corrected *p*-values < 0.05 (Appendix A). Inspection of these data identifies 24 loci with sensitivity ≥ 50 (a measure of the frequency of presence in isolates with the trait), but only 10 of those loci had a specificity score ≥ 50 (a measure of how specific the gene is in isolates with the trait). Of those 10 loci with higher specificity, five were annotated as hypothetical proteins and another as transposase. The other four loci encode functions in vitamin B12 import, ascorbate uptake, hexulose conversion, and transcriptional regulation of glucosamine utilization. We interpret these Roary–Scoary results to indicate that there is no evidence of specific gene acquisitions that are shared by the majority of sepsis isolates. Therefore, either there are no specific gene acquisitions that can convert a BCO pathogen into a sepsis pathogen, or individual clades have specific independent gene sets that must be acquired for transition to a sepsis pathogen based on their particular pan-genome composition.

To assess clade-specific gene acquisition, we used Scoary to analyze gene presence/absence for sepsis and BCO isolates in cluster 1 comprising French isolates (trait CIRMBP Cluster 1 in Appendix A). We compared four sepsis isolates to 22 BCO isolates within this clade. The Scoary results (Appendix A) identified only seven genes, with six annotated as hypothetical proteins and one as a transposase. None of these loci had significant corrected *p*-values for association but showed high specificity scores. Five of the loci were clustered in at least two of the sepsis isolate genomes and Phaster results showed that this region is likely a prophage (contig 1 from 109,451–157,474). Thus, our analyses failed to reveal compelling evidence for the gene acquisition driving the emergence of sepsis.

### 3.5. Are Core Genome Mutations Central to Sepsis or Disease Traits?

For the transition from BCO pathogen to sepsis pathogen, an alternative to gene acquisition through horizontal transfer is that the transition involves the mutation of resident genes. Evidence from sequence analysis of ancient Marek’s Disease Virus strongly supports that this virus has become more pathogenic, not by gene acquisition or gene rearrangement but through point mutations [38]. The emergence of sepsis through point mutations would also be consistent with the polyphyletic origins of sepsis isolates (Figure 1). Existing pathogens or commensals could experience a few key mutations that would allow them to survive in the blood and colonize organs.

We first used ParSNP to identify and determine the core genome SNPs that were differentially represented in BCO vs. sepsis isolates. The ParSNP analysis was limited to genomes known to be from cases of chicken disease (scored as 1 in chicken disease; Appendix A). For this analysis, we employed the same reduced representation from Cluster A (excluding NA isolates as in Sepsis Strict Reduced; Appendix A). We compared 17 sepsis isolates to 113 BCO isolates using the finished/complete genome BCO isolate SA1 as the reference. Since the reference genome was a BCO isolate, only SNPs over-represented in sepsis should represent mutations favoring sepsis. The core genome SNPs (*n* = 78,150) were filtered for those present in >94% of Sepsis Strict Reduced isolates (*n* = 967) and then for frequencies ≥0.30 in the sepsis isolates relative to the BCO isolates. This identified 34 SNPs: 32 bi-allelic and 2 tri-allelic (Appendix A). These 34 potentially diagnostic SNPs occurred in only six genes. Eight SNPs were in the first or second codon base positions and most likely to be missense, whereas 26 were in the wobble position and were more likely to be silent. Interestingly, four of the genes (Smc, YaaA, hypothetical protein, and AldC) appeared to be clustered, as they represented annotated PEG 229, 230, 231, and 234.

We analyzed the same reduced representation of chicken disease genomes using Maast as an alternative to ParSNP because the two programs use different alignment algorithms to genotype SNPs and are known to produce different results [35]. The reference genome was again the BCO isolate SA1. The maximum SNP genotype frequencies for Sepsis genomes from Maast were 0.54 (Appendix A), which was considerably lower than the 1.00 SNP frequencies identified by ParSNP (Appendix A). The Maast genotype data were filtered for SNPs > 50% in Sepsis (*n* = 252) and frequency difference (sepsis–BCO) ≥ 0.20, resulting in a total of 77 SNPs (74 biallelic and 3 triallelic), 7 intergenic, and 70 in coding sequences for 51 protein-encoding genes (Appendix A). Fifteen of the genes that flanked or spanned the SNPs were annotated as hypothetical proteins, but none of these hypothetical genes was PEG 231 identified in the ParSNP analyses. Of the four clustered genes from the ParSNP data, only the Smc gene was also identified in the Maast analyses. Both the ParSNP and Maast identified SNPs in the PTS system fructose-specific EIIABC component. The potential clusters of SNPs identified in the Maast analyses included those affecting PEG 346 to 373, 570 to 577, 629 to 637, and 744 to 753. There was also the potential for clustering genes affected by seven SNPs affecting four genes from PEG 594 to 603.

We extracted and aligned the encoded polypeptides for the six genes from ParSNP and 12 additional genes from Maast to identify all the variant polypeptide positions (Appendix A). The analyses included predicted polypeptides from all chicken disease isolates and included those from Cluster A that had been excluded from the initial ParSNP and Maast analyses. Each variant position was scored for frequency in the Sepsis Strict group of isolates relative to the BCO isolates. This identified 114 polypeptide variants where the Sepsis percentage was greater than 20 points higher than the percentage in BCO (green highlight in Appendix A). The highest frequency protein variations were the R555Q and A590V substitutions in the PTS system fructose-specific EIIABC, which were found in 75% of sepsis isolates but only in 33 and 29%, respectively, of BCO isolates (Appendix A). All 114 polypeptide variants were also scored for frequency in the Sepsis Strict Reduced (to reduce the influence of the higher number of sepsis isolates from Cluster A). Of the 114 protein variants, 15 were ≥50% in Sepsis Strict Reduced in seven genes (highlighted in yellow in Appendix A). These 15 SNPs are summarized in Table 3 and are candidates for key mutations driving the adaptation from BCO to sepsis.

The annotation of these seven genes and the literature support their possible roles in the virulence of *E. cecorum*. The hypothetical protein PEG231 was identified in the ParSNP analysis (Appendix A). BLASTp analysis at https://www.uniport.org (accessed December 2023) identifies this as a serine aminopeptidase S33 domain-containing protein. Hip1 in *Mycobacterium tuberculosis* is a S33 serine aminopeptidase that inhibits macrophage and dendritic cell functions [39]. PTS (phosphotransferase transport system) fructose-specific EIIABC has been found to regulate virulence expresssion and stress response in *Lysteria monocytogenes* [40], and biofilm and endocarditis in *Enterococcus faecium* [41]. CBCL1 orthologs in *Pseudomonas aeruginosa* are involved in the production of quorum-sensing signals [42]. BLASTp at uniprot.org identifies ElaA as a GNAT family N-acetyltransferase, which is a family of proteins involved in bacterial adaptation to diverse habitats [43]. EttA regulates protein synthesis in energy-depleted cells and is critical for bacterial survival during the long-term stationary phase [44]. RpoN encodes a sigma factor, σ^54^, for directing RNA polymerase to alternative promoters. Orthologs have been shown to regulate virulence determinants, including pili, flagella, and type III secretion systems (reviewed in [45]). YheH is known to function in the signaling pathway for sporulation in *Bacillus subtilis* [46]. Therefore, the genes identified have to do with environmental response and gene regulation. Further, there are at least an additional 39 genes from the Maast analysis that we have yet to assess whether they have coding variations differentially present in sepsis vs. BCO isolate genomes (Appendix A). There are also intergenic region SNPs that could affect the regulation of flanking genes.

## 4. Conclusions

Our analyses of all publicly available genomes of *E. cecorum* support a polyphyletic origin of pathogenicity in chickens as well as the recent outbreaks of early on-set sepsis in broiler flocks. This contrasts with previous works suggesting a common lineage in chicken pathogenesis [16]. Our analysis of the pangenome of *E. cecorum* identified no gene or gene clusters essential for the switch from chicken commensal to chicken pathogen or for chicken sepsis isolates. The cluster of 12 capsule-related genes identified by Borst et al. is highly prevalent in osteomyelitis and sepsis isolates, but it is not essential for the pathogenesis nor diagnosis of pathogenic isolates (Table 2). Analysis of core genome SNPs identified missense mutations in a subset of genes associated with osteomyelitis, which are further enriched in sepsis isolates. These genes may be involved in the bacterial response to stress, stationary phase survival, and cell-mediated immunity of the host. However, further research is warranted to confirm and expand these initial findings.

Future work should focus on obtaining additional isolates from sepsis outbreaks to determine how much pathogen genomic diversity is present in each outbreak and to assess whether the same genotypes are present in successive outbreaks in a facility. Previous works have failed to identify vertical transmission of *E. cecorum* using standard culture methods [10,18]. More sensitive DNA-based methods need to be applied to breeder hens, hatcheries, embryos, and broiler houses [47]. The virulence testing of the isolates is also problematic. Some have relied on chicken embryo lethality assays for the evaluation of *E.cecorum* isolates [17,48,49], but the direct relevance to pathogenicity post-hatch has been a concern [50]. Experimental infections of young chicks with isolates of *E. cecorum* have shown some promise [10] but have not been used to evaluate the relative virulence of different isolates. Some insects have been used in the virulence assays of human isolates of other Enterococcus species (reviewed in [51]). Alternative models could be used to discern whether sepsis isolates and BCO isolates have measurable differences in pathogenesis. Further, many of the genomes in national databases are poorly documented for specific clinical sources and host disease states; thus, there is a need for expanding the available genomes from well-documented cases.

## Figures and Tables

**Table 1 microorganisms-12-00250-t001:** Details of the *Enterococcus cecorum* strain source, date, location, and bird age. For each strain, the genome assembly in the number of assembled contigs, total assembly in base pairs, and NCBI BioSample are presented. For each DNA, the BLASTp results (average for percent identity × query coverage for protein genes and product genes 312 to 323) were used for the Borst virulence gene cluster. PopPUNK cluster shows the groupings based on phylogenomic comparisons. Not available: n/a.

Strain	Source	Date Collected	Collect Location	Bird Age	Assembly Contigs	Total bp	BioSample	BLASTp	PopPUNK Cluster
1415	Tibial pus	6/23/2016	farm16	4.3	65	2,411,963	SAMN38750234	100	A
1675	Femoral head necrosis	11/26/2020	UAPRF	8	56	2,188,186	SAMN38750235	100	B
1749	Liver	n/a	farm15	0.5	93	2,494,030	SAMN38750236	15	solo
1750	Heart	n/a	farm22	3	49	2,281,603	SAMN38750237	100	C
1751	n/a	n/a	farm19	2.5	49	2,300,397	SAMN38750238	100	A
1752	Liver	n/a	farm8	3.6	32	2,279,238	SAMN38750239	17	solo
1753	Heart	n/a	farm15	3.1	51	2,240,969	SAMN38750240	100	A
1754	n/a	n/a	farm2	n/a	47	2,180,738	SAMN38750241	100	A
1755	Heart	n/a	farm2	n/a	58	2,366,287	SAMN38750242	100	A
1756	Heart	n/a	farm8	3.2	62	2,288,864	SAMN38750243	100	A
1757	Heart	n/a	farm4	3.3	53	2,297,190	SAMN38750244	100	A
1758	Heart	n/a	farm22	3	53	2,249,959	SAMN38750245	100	A
LB5753	Heart	2/5/2021	farm17	3	56	2,226,311	SAMN38750246	100	A
LB5754	Heart	2/12/2021	farm9	5	62	2,234,879	SAMN38750247	100	A
LB5756	Heart	3/8/2021	farm13	2.3	62	2,239,264	SAMN38750248	100	A
LB5759	Heart	3/8/2021	farm11	2.6	75	2,392,627	SAMN38750249	100	A
LB5800	Heart	3/11/2021	farm3	3	42	2,212,442	SAMN38750250	100	B
LB5836	Vertebral osteomyelitis	9/29/2020	farm6	12	69	2,320,142	SAMN38750251	100	C
LB5840	Heart	10/13/2020	farm5	n/a	57	2,258,228	SAMN38750252	100	A
LB5843	Vertebral osteomyelitis	10/20/2020	farm7	8	70	2,319,059	SAMN38750253	100	C
LB5856	Heart	12/21/2020	farm14	n/a	81	2,378,014	SAMN38750254	40	solo
LB5857	Egg transfer residue	3/26/2021	hatchery 1	Eggs	47	2,180,187	SAMN38750255	100	B
LB5860	Cull eggs	3/24/2021	hatchery 1	Eggs	102	2,230,151	SAMN38750256	100	D
LB5864	Heart	12/9/2020	farm20	2.3	61	2,239,173	SAMN38750257	100	A
LB5872	Heart	4/21/2021	farm18	n/a	65	2,365,322	SAMN38750258	100	A
LB5876	Egg transfer residue	5/10/2021	hatchery 1	Eggs	37	2,183,317	SAMN38750259	100	B
LB5880	Air sacculitis	4/29/2021	farm21	2	38	2,146,928	SAMN38750260	100	B
LB5916	Egg transfer residue	6/4/2021	hatchery 1	Eggs	40	2,193,104	SAMN38750261	100	B
LB5922	n/a	6/16/2021	farm12	1	88	2,169,308	SAMN38750262	99	D
LB5923	intestinal tract	6/21/2021	farm 1	1.3	58	2,239,406	SAMN38750263	100	B
LB5924	Air sacculitis	4/2/2021	farm10	2.4	61	2,296,440	SAMN38750264	100	A
LB5925	Air sacculitis	4/2/2021	farm10	2.4	61	2,297,136	SAMN38750265	94	A
LB5926	Egg transfer residue	4/16/2021	hatchery 2	Eggs	103	2,277,367	SAMN38750266	100	D
LB5927	Egg transfer residue	4/16/2021	hatchery 2	Eggs	105	2,276,597	SAMN38750267	97	D

**Table 2 microorganisms-12-00250-t002:** BLASTp scores for protein-encoded gene products from the Borst gene cluster, based on strain categorization by phenotype rather than chicken disease (none), chicken disease (CD), bacterial chondronecrosis with osteomyelitis (BCO), or strict sepsis (SS). See the results for definitions of phenotype categorization. For each protein-encoded gene, the value is the average for that phenotype of the BLASTp scores for (percent identity × query coverage)/100. For each phenotype, the count is the number of strains in that phenotypic category. The average ± SEM was computed for all 12 polypeptides. Individual gene scores for each isolate are provided in Appendix A.

Phentotype	Count	Borst Gene Cluster Protein Encoded Gene												Average ± SEM
		312	313	314	315	316	317	318	319	320	321	322	323	
None	82	53	35	37	49	30	43	31	31	45	43	39	91	44 ± 5
CD	145	85	77	79	83	75	79	74	75	84	85	82	98	81 ± 2
BCO	113	84	76	77	82	73	77	72	73	81	83	79	97	80 ± 2
SS	34	88	80	83	86	81	83	79	79	89	88	88	98	85 ± 2

**Table 3 microorganisms-12-00250-t003:** Summary of fifteen polypeptide variants most prevalent in *Enterococcus cecorum* sepsis isolates. Entries for each variant include the gene name, predicted function, amino acid positions (AA Pos) affected, the reference residues (Ref), substitution (Alt), and percentage of isolates in each of the three phenotypic trait groups.

Gene					Phenotypic Trait Group		
Name	Function	AA Pos	Ref	Alt	BCO	Sepsis Strict	Sepsis Strict Reduced
PEG231	Serine aminopeptidase S33 domain-containing protein	261	T	A	35	66	50
EIIABC	PTS fructose-specific EIIABC component	555	R	Q	33	75	64
		590	A	V	29	75	64
CBCL1	4-chlorobenzoate--CoA ligase	244	G	H	43	69	55
		246	I	L	43	69	55
		249	H	Y	43	69	55
ElaA	Gcn5-related N-acetyltransferase (GNAT family)	128	N	K	34	69	55
EttA	Energy-dependent translational throttle protein	106	S	A	40	69	55
		534	T	A	39	66	50
RpoN	RNA polymerase σ54 factor	12–13	TQ	--	25	38	50
		22	T	S	26	38	50
YheH	putative multidrug resistance ABC transporter ATP-binding/permease protein	13	I	L	43	72	59
		578	D	N	28	66	50
		590	S	I	28	66	50
		592–593	EEI	GAD	28	69	55

## Data Availability

All genomes analyzed are available in the databases at the National Center for Biotechnology Information (https://www.ncbi.nlm.nih.gov).

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
