# Peer review of "Molecular Genomic Analyses of Enterococcus cecorum from Sepsis Outbreaks in Broilers"

_microorganisms, 2024, doi:10.3390/microorganisms12020250_

Round 1
Reviewer 1 Report
Comments and Suggestions for Authors
This research studied molecular genomic analyses of Enterococcus cecorum from sepsis outbreaks in broilers, which seems to be valuable and interesting. However, there are several questions to be solved.
Questions:
1. The references are too old, please update them with the new ones.
2. Please add the conclusion part for this manuscript.
3. Have you got the permission of the online data?
4. Please add continuous line number for this manuscript.
5. Authors should compare their findings with related research.
6. For the abstract part, please define the SNP but not use it without definition.7. About the keywords, please delete some words such as osteomyelitis and pathogenesis which may not be closely related with the research.
8. What conclusion do you obtain?
9. What's the meaning of the research?
Comments on the Quality of English LanguageThe English is OK.
Author Response
Questions:
- The references are too old, please update them with the new ones.
Response: The reviewer has not specified which “old” references are no longer allowed. The breakdown of our citations is below which shows that the vast majority are from the past 10 years with only a few which are the proper citations for historical reference to the E. cecorum. Nearly 20% of our citations are from 2022 to the present.
|
Year |
Count |
|
2024 |
1 |
|
2023 |
5 |
|
2022 |
4 |
|
2021 |
3 |
|
2020 |
2 |
|
2019 |
1 |
|
2018 |
1 |
|
2017 |
5 |
|
2016 |
4 |
|
2015 |
3 |
|
2014 |
7 |
|
2012 |
2 |
|
2011 |
1 |
|
2009 |
1 |
|
2008 |
2 |
|
2006 |
2 |
|
2005 |
2 |
|
2004 |
1 |
|
2003 |
1 |
|
1992 |
1 |
|
1989 |
1 |
- Please add the conclusion part for this manuscript.
Response: We have added a section title for the conclusions and expanded them.
- Have you got the permission of the online data?
Response: The data in our manuscript is based on 32 genomes we sequenced and/or assembled. The other genomes were downloaded from NCBI databases. That means they are in the public domain and we do not need any permission.
- Please add continuous line number for this manuscript.
Response: Our submission did have line numbers but the MDPI rendering software appears to have removed them. That is not under our control. We added them back to the revision in the template version that MDPI generated.
- Authors should compare their findings with related research.
Response: We have compared our data to that of Borst et al. and Laurentie et al., which are the only two previous comparative genomics works on this organism. Those publications are highlighted in the Introduction and then we contrast our results with both in the Results and Discussion.
- For the abstract part, please define the SNP but not use it without definition. :
Response: We have done this and then defined the acronym in the methods.
- About the keywords, please delete some words such as osteomyelitis and pathogenesis which may not be closely related with the research.
Response those two words belong in the keywords because we are comparing sepsis isolates to osteomyelitis isolates, and looking for genetic determinants that might contribute to pathogenesis.
- What conclusion do you obtain?
Response: the conclusions section has been identified and expanded.
- What's the meaning of the research?
Response: the meaning is covered in the results and discussion section, and the significance framed in the conclusions. We described the purpose or focus of the research and framed it with respect to this new industry problem in the Introduction.
Reviewer 2 Report
Comments and Suggestions for Authors
This manuscript is very interesting and fits the scope of the journal. The quality of the presentation needs a major revision. Please address the following issues:
1-No line numbers in this version. Please, address this issue in the revised version of the manuscript.
2-Abstract: Please edit it again to represent the content of the study. Please organize the abstract section into rationale, objectives, methodologies, results, and conclusion.
3-Introduction: This section is fine. But please state the breed/species, especially in the last paragraph of this section.
4-Circumstanses of E. cecorum isolates should be clarified in detail in this study.
5-I wonder whether the authors got approval for the experiment (ethical approval certificate).
6-It would be better to divide the material and methods section into several subheadings. What about the analysis and statistics?
7-Conclusion section was missed.
8-I wonder about the cost-benefit of the experimental protocol. How many breeders or companies would invest in this type of research? Argue and justify the choice of Enterococcus cecorum!
Author Response
1-No line numbers in this version. Please, address this issue in the revised version of the manuscript.
Response: Our submission did have line numbers but the MDPI rendering software appears to have removed them. That is not under our control. We added them back to the revision in the template version that MDPI generated.
2-Abstract: Please edit it again to represent the content of the study. Please organize the abstract section into rationale, objectives, methodologies, results, and conclusion.
Response: a review of the author guidelines and recent publications in Microorganisms does not indicate that the abstract must be divided into subsections.
3-Introduction: This section is fine. But please state the breed/species, especially in the last paragraph of this section.
Response: breed/species of broilers? We have added broilers in key places and specified E. cecorum more frequently in the Introduction.
4-Circumstanses of E. cecorum isolates should be clarified in detail in this study.
Response: We have provided the details on the isolates we characterized in Table 1. Because corporate entities provided these samples we were required to anonymize the locations because of corporate privacy concerns and market liabilities. Our lawyers have agreed that the companies be protected.
5-I wonder whether the authors got approval for the experiment (ethical approval certificate).
Response: there are no animal experiments, and we do have APHIS registration for receipt and shipment of the cultures. We are not certain that needs to be included in the manuscript.
6-It would be better to divide the material and methods section into several subheadings. What about the analysis and statistics?
Response: methods has been subdivided. The analyses and statistics are all there in tables and the results.
7-Conclusion section was missed.
Response:
8-I wonder about the cost-benefit of the experimental protocol. How many breeders or companies would invest in this type of research? Argue and justify the choice of Enterococcus cecorum!
Response: The funding is listed in the manuscript and came from State of Arkansas resources. The question as to whether breeding or integrators would invest in this research is outside the scope of our research. The introduction, results and discussion all detail the reasons for working on E. cecorum given a newly emerging issue with sepsis associated with this organism. Unfortunately, the magnitude of the problem is not in the public domain and is still only known to the commercial companies. Given that corporate reputations and market share are critical economic drivers, the only information we get is “off the record.”
Reviewer 3 Report
Comments and Suggestions for Authors
The paper could have a certain interest; however, I have many concerns. No line numbers are present and this makes very difficult to precisely address the comments.
The section “Results and discussion” is very long and confusing. I would suggest to separate this section in two: “results” were an accurate and clear description of the results is reported, also avoiding to report materials and methods description as occurs in the first part of this section; “discussion” were an analysis of the results and a comparison with similar results present in the scientific literature, is provided.
Also, materials and methods section should be divided into section (e.g. “strains” accurately describing the number and the features of strains included in the study) and “sequencing”.
With this in mind, I cannot consider the paper acceptable for publication.
Other comments
Check nomenclature of microorganisms (e.g. in the introduction section, “Enterococus” is reported).
Comments on the Quality of English LanguageI wuold suggest a revision by a native speaker.
Author Response
The paper could have a certain interest; however, I have many concerns. No line numbers are present and this makes very difficult to precisely address the comments.
Response: Our submission did have line numbers but the MDPI rendering software appears to have removed them. That is not under our control. We added them back to the template version that MDPI generated.
The section “Results and discussion” is very long and confusing. I would suggest to separate this section in two: “results” were an accurate and clear description of the results is reported, also avoiding to report materials and methods description as occurs in the first part of this section; “discussion” were an analysis of the results and a comparison with similar results present in the scientific literature, is provided.
Response: We chose to combine the results and discussion as the analyses proceeded in a scripted fashion with the phylogenomics, the analysis for gene acquisition, and then the core genome mutations. As such we needed to discuss the implications of each finding with regard to previous research. Therefore, there is some discussion warranted before moving to the next part of the results.
Also, materials and methods section should be divided into section (e.g. “strains” accurately describing the number and the features of strains included in the study) and “sequencing”.
Response: We have done this
With this in mind, I cannot consider the paper acceptable for publication.
Other comments
Check nomenclature of microorganisms (e.g. in the introduction section, “Enterococus” is reported).
Response: we are not certain about the reviewers issue. For listing the genus only it is not italicized.
Comments on the Quality of English Language
I wuold suggest a revision by a native speaker.
Response: the first and second authors are both native English speakers, and the last author is fluent in four languages, one of which is English. We are perplexed by this comment from the reviewer.
Reviewer 4 Report
Comments and Suggestions for Authors
Comments to the Authors of manuscript number: microorganisms-2836080 entitled “Molecular genomic analyses of Enterococcus cecorum from sepsis outbreaks in broilers”.
The text examines Enterococcus cecorum in broiler sepsis outbreaks, attributing its origin to core genome mutations rather than gene acquisition. The species, part of avian intestinal flora linked to osteomyelitis, is now associated with sepsis in broilers. Phylogenomic analyses show close relations between sepsis isolates and other strains, with no distinct gene acquisitions. Core genome SNP analyses identify mutations supporting a mutational origin for sepsis isolates.
1. “nature” is too wide meaning.
2. The introduction lacks specific citations for certain claims, such as the association of Enterococcus faecalis, Enterococcus faecium, and Enterococcus avium with infections in humans. Including precise citations would enhance the credibility of the information.
3. The text mentions recent sepsis outbreaks caused by E. cecorum but doesn't provide details on the scale, impact, or specific locations of these outbreaks. Including such information would add context and help readers understand the severity of the issue.
4. The text mentions a cluster of genes distinguishing osteomyelitis isolates and the potential involvement of capsular modifications in virulence. However, it doesn't delve into the significance of these findings or their implications for addressing the issue. Adding a brief explanation would enhance the reader's understanding.
5. The text mentions questions to be addressed but doesn't provide a clear outline of the study's methodology or a roadmap for how these questions will be investigated. Including a brief section on the planned approach or methodology would add clarity.
6. there is no any hypothesis
7. The text mentions that E. cecorum isolates were obtained from the University of Arkansas Poultry Diagnostic Laboratory. However, it lacks information on the number of isolates and whether they were from different outbreaks or regions. Providing more details on the isolate source would enhance the study's transparency.
8. Although the DNA isolation and sequencing methods are outlined, specific details such as the DNA concentration, quality assessment, and the sequencing platform used (aside from Illumina) are not mentioned. These details are crucial for understanding the reliability of the genomic data.
9. Results and discussion:
a. The section is highly detailed, which is good for a scientific paper. However, consider breaking down the information into smaller paragraphs for easier readability.
Utilize bullet subheadings to emphasize key findings and make the content more digestible.
b. The phylogenomic analysis using PopPUNK and ParSNP is well-described. However, for clarity, consider providing a brief overview of the key findings before delving into details. This helps readers understand the context.
c. Provide a brief rationale for the choice of specific tools and parameters. Why were PopPUNK, ParSNP, Scoary, and Maast chosen? This explanation would enhance the transparency of your methodology.
d. The expansion of the pangenome compared to previous studies is a key finding. Clarify whether this expanded pangenome suggests increased genomic diversity or if there are functional implications.
e. The discussion about gene acquisition and mutation is intriguing. Elaborate more on the implications of these findings for understanding the emergence of sepsis.
f. The identification of potential key mutations is a significant result. Clearly state the implications of these mutations and how they may contribute to the transition from BCO to sepsis.
10. The section discussing future work is insightful. Consider expanding on how the suggested experiments or analyses would contribute to a deeper understanding of E. cecorum virulence.
11. the clear conclusion is needed
Author Response
- “nature” is too wide meaning.
Response: we used nature because it has been detected in aquatic systems, soil, plants, etc. The genus does not appear to be restricted to animals, or even organisms.
- The introduction lacks specific citations for certain claims, such as the association of Enterococcus faecalis, Enterococcus faecium, and Enterococcus avium with infections in humans. Including precise citations would enhance the credibility of the information.
Response: There are four citations given for these statements. “Some species are found in diverse vertebrates, such as Enterococcus faecalis, Enterococcus faecium, and Enterococcus avium, which have been isolated from both mammals and aves [1; 2]. In humans these species have been associated with infections of the urinary tract, blood stream, heart, and central nervous system [7; 8].”
- The text mentions recent sepsis outbreaks caused by E. cecorum but doesn't provide details on the scale, impact, or specific locations of these outbreaks. Including such information would add context and help readers understand the severity of the issue.
Response: we appreciate the issue/desire of informing readers about the extent of this issue. However, the issue is in the industry where corporate standings and reputations are critical. Growers and integrators may share anecdotal information, but they do not wish to “go on the record” or divulge animal/economic losses. They have privately shared their concerns and asked for research to inform their decisions. As such, we do not know what the actual losses are or how many integrators are having issues. For the same reason we can not share the actual locations or companies which shared isolates and information. That is why all isolates had to be anonymized in Table 1.
- The text mentions a cluster of genes distinguishing osteomyelitis isolates and the potential involvement of capsular modifications in virulence. However, it doesn't delve into the significance of these findings or their implications for addressing the issue. Adding a brief explanation would enhance the reader's understanding.
Response: the cluster of 12 genes is mentioned in the introduction and then we analyze this cluster across all 227 genomes with respect to pathogen trait in Table 2. We have added a repast to the Conclusions.
- The text mentions questions to be addressed but doesn't provide a clear outline of the study's methodology or a roadmap for how these questions will be investigated. Including a brief section on the planned approach or methodology would add clarity.
Response: we have added some text at the end of the introduction as a prequel.
- there is no any hypothesis
Response: although the word hypothesis was not used the purpose and goal was stated near the end of the introduction: “Questions we wished to address included: whether specific mutations, or gene acquisition are driving recent sepsis outbreaks, whether sepsis and osteomyelitis isolates are distinct lineages, and whether high resolution bacterial genomics can inform responses to sepsis outbreaks.”
- The text mentions that E. cecorum isolates were obtained from the University of Arkansas Poultry Diagnostic Laboratory. However, it lacks information on the number of isolates and whether they were from different outbreaks or regions. Providing more details on the isolate source would enhance the study's transparency.
Response: to avoid confusion we have deleted that sentence at the start of methods. The sources and dates of isolation are presented in Table 1, which is referenced at the start of the Results and Discussion.
- Although the DNA isolation and sequencing methods are outlined, specific details such as the DNA concentration, quality assessment, and the sequencing platform used (aside from Illumina) are not mentioned. These details are crucial for understanding the reliability of the genomic data.
Response: We would argue that adding the DNA amounts purified would be superfluous to the quality of the assemblies. The sequence reads and assemblies have all been deposited in NCBI and biosample IDs, contig count, and total assembly size in bp are provide in Table 1. We have added the platform used by SeqCenter in methods. As described in the methods some of the assemblies were made with sequence reads provided to us by Dr. Borst at NCSU. We have added that those data are 2x251 data
- Results and discussion:
- The section is highly detailed, which is good for a scientific paper. However, consider breaking down the information into smaller paragraphs for easier readability.
Utilize bullet subheadings to emphasize key findings and make the content more digestible.
Response: we have added the subsections
- The phylogenomic analysis using PopPUNK and ParSNP is well-described. However, for clarity, consider providing a brief overview of the key findings before delving into details. This helps readers understand the context.
Response: The summary is provided. In the revised document see Lines 181-185, then there is a “walk-through” for several of the clusters (lines 185-206). We added some paragraph breaks to improve readability.
- Provide a brief rationale for the choice of specific tools and parameters. Why were PopPUNK, ParSNP, Scoary, and Maast chosen? This explanation would enhance the transparency of your methodology.
Response: we provided the software we chose to use and why we used both ParSNP and Maast. There are others who would have chosen to use other software and we have attempted to cover the limitations on our choices. If we digressed into a debate over the merits of different software the manuscript would become a methodology paper which would detract from the scientific discourse.
- The expansion of the pangenome compared to previous studies is a key finding. Clarify whether this expanded pangenome suggests increased genomic diversity or if there are functional implications.
Response: The differences in our pangenome numbers and those from INRAE probably relate to our using twice as many genomes, and our use of equal numbers from France and the USA. The INRAE samples were almost all from their collections so less diverse. We are not experts on pangenomics but experts might be able to clarify the implications. However, we don’t want to put speculation into the mix in the results.
- The discussion about gene acquisition and mutation is intriguing. Elaborate more on the implications of these findings for understanding the emergence of sepsis.
Response: this is now covered further in the Conclusions section
- The identification of potential key mutations is a significant result. Clearly state the implications of these mutations and how they may contribute to the transition from BCO to sepsis.
Response: the discussion of the role of the identified polypeptides is presented in the last paragraph of the Results and Discussion section. To go further would require presentation of structural data and mapping mutations onto those structures. Given the number of genes that will constitute much additional work and can be the basis for further experiments. However, as we state in the conclusions more data is needed regarding the genomic diversity within and between outbreaks along with good sample data for disease state. We need that data before focusing on individual mutations and their roles.
- The section discussing future work is insightful. Consider expanding on how the suggested experiments or analyses would contribute to a deeper understanding of E. cecorum virulence.
Response: We have expanded the conclusions
- the clear conclusion is needed
Response: We have expanded the conclusions
Round 2
Reviewer 2 Report
Comments and Suggestions for Authors
Thank you very much for your revision. Please address the appended comments:
1-Please edit the abstract section. Simply, state the methods and conclusion.
2-State that Animal Ethical Approval was not provided in this study (mention the reason).
Author Response
Reviewer 2 comments
1-Please edit the abstract section. Simply, state the methods and conclusion.
Response: we have reviewed the abstract organization and content. We do not wish to delete the two sentences that provide the context for our research (the recent sepsis outbreaks and the age of onset). We mention the methods used (phylogenomics, pan genome, and core genome analyses). We state the results of those analyses. If we were to include the conclusions this would greatly expand the abstract length which does not seem to be the intent of the reviewer’s comment. Further, most of the conclusions are admittedly speculative. Therefore, we prefer to stay with the abstract as written.
2-State that Animal Ethical Approval was not provided in this study (mention the reason).
Response: We have added a section which hopefully explains why we needed no IACUC approvals.
Reviewer 3 Report
Comments and Suggestions for Authors
I have not any more comments.
Comments on the Quality of English LanguageEnglish just need some minor correction.
Author Response
Reviewer 3 comments:
English just need some minor correction.
Response: we found a few places where we added additional punctuation, or rearranged the words to be clearer. Those are marked with track changes.